# Smart Sensing Period for Efficient Energy Consumption in IoT Network

**DOI:** 10.3390/s19224915

**Published:** 2019-11-12

**Authors:** Woojae Kim, Inbum Jung

**Affiliations:** Department of Computer Information and Communication Engineering, Kangwon National University, Chuncheon, Gangwondo 200-701, Korea; kkkwo1020@naver.com

**Keywords:** internet of things, smart sensing period, neural network, power consumption rate, battery energy

## Abstract

The devices included in IoT networks have sensors and actuators for monitoring their surroundings. These operate on battery energy, according to the characteristics of the environment in which they are deployed. To enhance the longevity of IoT networks, the devices need to avoid any unnecessary sensing operations in order to reduce the power consumption rate. However, as existing sensing methods use a fixed sensing period policy, battery power wastage is inevitable. In this paper, a smart sensing period policy is proposed for efficient energy consumption in an IoT network. The proposed method uses a learning model based on a back-propagation neural network. Within the target time, it can efficiently use the battery energy without any surplus or wastage in the quantity of preserved battery energy. In experiments, our proposed method shows improved results in battery energy consumption rates compared to the existing sensing period methods.

## 1. Introduction

IoT technologies are being actively developed, and already provide many services to people. In an IoT network, the system devices are interconnected—they interact with each other and exchange collected sensor data. However, working environment factors such as different operating systems, communication protocols and hardware can present numerous challenges to the development of IoT service applications. IoT middleware platforms have recently been proposed and researched to help solve this [1,2,3,4]. In middleware platforms, the integrated development environment is provided, and furthermore the response time between devices can be reduced through resource management functions. These middleware platforms aim to reduce power consumption, since the devices in IoT (Internet of Things) usually run on battery power. However, sensing operations currently use fixed sensing period policies. Even if a variation in data collection does not appear, the devices perform sensing operations according to a predetermined sensing period, which wastes battery energy carrying out unnecessary sensing processes.

The smart devices in IoT networks include sensors, actuators, power modules, device platforms, communication modules and so on. They operate using the energy of an embedded battery. The operational characteristics of these devices therefore have a large impact on the energy consumption rate. Unlike a wireless sensor network with a unique sink node, the devices included in an IoT network communicate with each other without restriction; thus, a greater variety of IoT services can be provided for people. To achieve this, the devices should be able to transmit to, or exchange collected data with, relevant neighbor devices. Since sensors and transmitter and actuator devices are used in this process, the battery power consumption is severe. The exhaustion of battery power in devices has a negative impact upon the longevity of IoT networks, and consequently much previous research has focused on how to effectively utilize battery power, investigating the minimum power usage in sensing operations, low power communications, long-term battery usage without recharge or replacement, among other things [5,6,7,8,9].

The sensing methods of deployed devices have a large impact upon battery lifetimes in IoT networks, and an effective method of battery usage is subsequently needed to prolong the lifetime of IoT networks themselves. In this paper, a smart sensing period method is proposed for effective battery usage. The proposed method is based on aback-propagation neural network (BPNN). The smart sensing period method can dynamically control the sensing period until a target survival time is reached, hence can avoid a waste or surplus residue of battery power. To find the best sensing period for the specified working conditions, a learning model is designed using the BPNN. The model, in experiments, takes three properties as its learning data: the time remaining until the target time, the quantity of remaining battery power and the amount of power consumed by a single sensing operation. The proposed method deduces the best sensing period, through the learning model, for the current working environment of deployed devices.

To evaluate the performance of the proposed smart sensing period, the experimental environment is implemented on a Raspberry Pi. This device is usually used as a smart device in IoT, and is composed of a 1-GB memory and a quad CPU core. The related hardware and software are open source and available to the public, thus it has advantages in learning and application for various IoT service fields. Even if the Raspberry Pi device has better performance abilities than existing IoT devices, the iterative and complex computational work of a BPNN is still a burden to the Raspberry Pi. To avoid performance degradation, we design a small and light-weight model’s BPNN. In experiments, the smart sensing period method uses the battery power more effectively compared to the existing fixed sensing period methods. As a result, it can contribute to the longevity of IoT networks [10,11].

The rest of this paper is organized as follows. Section 2 describes work relevant to the IoT middleware platforms, existing sensing period methods and back-propagation neural network algorithm. In Section 3, the smart sensing period method is proposed based on the back-propagation neural network. In Section 4, the implementation environment for the smart sensing period is explained. In Section 5, the performance of the smart sensing period method and other fixed sensing period methods is evaluated. Section 6 concludes the paper and proposes future work.

## 2. Related Work

### 2.1. IoT Middleware

Various types of IoT middleware have been researched, to keep pace with 4th industrial revolution. For example, the *MOSDEN* is based on the field of wireless sensor networks, the *IoTivity* on distributed IoT middleware hosted by the Linux Foundation, and the *Californium* and *MinT* on multi-core platforms [12,13,14]. In particular, the *MinT* is an open source project that provides an effective application development platform for various IoT service fields. In this paper, the proposed smart sensing period control is implemented using this platform. The *MinT* provides abstracted interfaces for an IoT composed of heterogeneous devices, and as such has a convenient application development environment [15].

Figure 1 shows the *Resource Management Layer* for the *MinT*, which supervises the sensing data. The *Property Manager module* of Resource Management Layer controls the device sensors by applying a periodic or aperiodic sensing policy. In the aperiodic mode, the sensing operation is performed only when the client requests the sensing data. If the number of these requests increases, the time taken to process sensor data also increases, and the response of the device is delayed. In the periodic mode, once a fixed sensing period is set the sensor executes its operations according to the assigned period and saves the collected data. The saved sensor data is valid until the next sensing period commences. In the periodic mode, since the automatically saved sensor data can be transmitted to the clients, a constant response time is maintained. However, if the sensing period is set too long, the relatively recent sensor data cannot be delivered to the clients.

### 2.2. Sensing Period Control

In wireless sensor networks, dynamic sensing period methods use a *sleep* function to reduce the battery power consumption [16]. In these approaches, the *sleep* function operates dynamically on the period. It compares the current sensor data to the previous sensor data, if large variations are detected, the sensing period is reduced by half per period. On the other hand, if there are no variations in the collected data, the sensing period is increased by a 10% increment per period. The power consumption can be decreased by varying the increments. Since this approach allows the sensing period to be controlled sensitively according to the varying circumstances, the battery energy consumption rate is decreased to some extent. However, the degree of variation in data collection is not specialized according to sensor type. In addition, it does not consider the operational conditions that obtain when a sensor device works with multiple other sensors.

An adaptive sampling mechanism controls the data collection speed when each sensor collects information on its environment [17]. As the sampling data is compared to the environmental model, the error value can be calculated. If the error value is more than the target margin value, a higher sampling rate is applied in the data collection process. Otherwise, a lower rate is applied. The error value is calculated from the sampling data and internal model. This control mechanism is designed based on the sampling rate and the complexity of the control loop and internal model. As the complexity of the sampling control loop and the internal model increases, the sampling rate is changed to preserve accuracy. However, the more accuracy achieved, the larger the burden on computation, and the more additional power required.

### 2.3. Back-Propagation Neural Network

The back-propagation neural network (BPNN) uses a layered neural network; as such, it controls the connection strength of middle layers, enabling it to learn a diverse range of subjects [18]. To solve the linear inseparability problems, for instance the exclusive OR, of perceptron algorithms, the BPNN controls the connection strength of middle layers via the output error. The error is a unique criterion by which to measure the performance of the neural network. It uses the sum of the squared errors [19].

Figure 2 shows the back-propagation neural network structure. The input values for *Input Layer* are transferred into the *Hidden Layer* and continue to the *Output Layer*, and the output value y of the *Output Layer* is taken as the final output. The output y is compared to the correct answer for the learning data, and the error calculated. The error is then transmitted from the output side to input side. In this process, the total error is distributed to each neuron of middle layer according to the appropriate ratio. This process is performed repeatedly until the final output error is within the target error range.

During the learning process, the output of each neuron becomes the input of other neurons located in the direction of progress. In addition, these output values are used in the activation function. Several activation functions can be applied in neural network algorithms. The activation function is selected based on the characteristics of the learning data. *Sigmoid* or *ReLU* functions exist as the representative linearly inseparable functions. The *Sigmoid* function gives an output value between 0 and 1. The *ReLU* function has an output value of 0 only when the output is below 0, otherwise, the output value is identical to the input one.

## 3. Smart Sensing Period Control in IoT Networks

In existing studies, the sensing period is controlled on the basis of the amount of collected sensor data. If a sudden change of collected data volume occurs, the sensing period is shortened to adequately reflect the events. Otherwise, as the sensing period is set increasingly longer, the amount of data collected from sensors is reduced. However, this approach is not appropriate in an IoT environment, because the devices in IoT have a limited battery energy. A more energy efficient method is here needed to make decisions regarding the sensing period, because it has a great impact upon the longevity of the IoT network. In this paper, the BPNN algorithm is used to manage the effective battery usage of devices in the IoT network. This algorithm can solve the problem of linear inseparability. Since the batteries show non-linear usage characteristics dependent on the working environment, the characteristics of the back-propagation neural network algorithm can contribute to finding the smart sensing period in IoT. In addition, this algorithm can be easily modified into a light-weight neural network model, to relieve the computation burden in IoT devices.

The battery lifetime is not linearly proportional to power consumption. In the case where 1 Watt of power is used for a sensing process, if the battery capacity is 100 Watt and the sensing period 1 s, 100 sensing operations are possible in 100 s. However, the sensing power consumption is not always constant. External conditions can affect the battery lifetime unexpectedly. Therefore, the dynamic sensing period needs to reflect the current working status of the battery. In this paper, the back-propagation neural network method is applied to satisfy the dynamic sensing period requirements, and to reflect the non-linear energy consumption characteristics of the battery. Moreover, since the neural network algorithm is constructed with a simple software architecture model, it not only uses minimal computing resources but also reduces the power consumption rate. Based on this approach, the smart sensing period can be varied dynamically according to the amount of remaining battery power [20].

Figure 3 shows the flowchart for the smart sensing period algorithm. First, when the smart sensing period algorithm begins, total time and power values are inputted and form the initialization step. In ①, the time variable denotes the time left until the inputted total time, and the power variable represents the power remaining after the initialization step has been executed. After the initialization step, the timer is set running as the function timer( ) in ②. Second, if the remaining time and power values are not 0, the function BPNN( ) calculates the smart sensing period best suited to the current battery working conditions. The sensing period for devices is set with the calculated sensing period value. The sensing operation is performed once every sensing period, so that the battery power is reduced by the amount of energy required for a single sensing operation. These steps are repeated until the remaining power is 0. When this happens, the sensing period control is stopped. The procedures of smart sensing period control are detailed in the next subsections.

### 3.1. Sensing Period Calculation

To calculate the sensing period, the number of sensing operations available within 1 s is needed. In addition, the amount of power consumed by one sensing operation should be measured. For example, if the present sensing period is 100 ms, it means that 10 sensing operations are available within 1 s (1 s/100 ms = 10). With these two values, we can calculate the amount of power consumed per second, by multiplying the number of sensing operations per second by the power consumed in one sensing operation. If the power consumed in 1 sensing operation is 0.1 W, we know that 1 W of power is consumed per second (10 × 0.1 W = 1 W). Equation (1) shows the amount of power consumed in 1 s. The current period is denoted as *period*, and the power consumed per sensing operation is marked as *P_S_*. The consumed power per second is *P*. Equation (2) represents the total power (*P_T_*) needed to reach the target time. It is found from the consumed power per second and the remaining time (*T*). From Equation (2), we can derive the period shown in Equation (3). In Equation (3), for the initial step, *P_T_* and *T* are designated as the total power and total time. In the later steps, whenever the sensing period is updated, they are calculated using the remaining power and time.

Each sensor has an absolute minimum time required for 1 sensing operation. This time is composed of both the internal waiting time before a sensing operation commences and the processing time for sensed data. For this reason, we use millisecond units rather than seconds. Equation (4) shows the process to convert to milliseconds.
(1)$$\frac{1}{period}\times P_{S}=P$$
(2)$$\frac{1}{period}\times P_{S}\times T=P_{T}$$
(3)$$\frac{P_{S}\times T}{P_{T}}=period$$
(4)$$period\left(s\right)\times 10^{3}=period\;\left(ms\right)$$
period = current period; $P_{S}$ = one-time sensing power consumption; $P_{T}$ = total power; T = remaining time.

### 3.2. Learning Data Configuration

Table 1 shows part of the learning data collected with an ultrasonic sensor. The learning data is composed of four quantities: power consumed in one sensing operation, remaining time, remaining power and period. Since the amount of power consumed in a sensing operation differs between companies, the data sheets provided by the manufacturers are used. The remaining time and power are determined by the user inputs. In this paper, 100 s and 40 W are chosen as the parameters for the experiments.

### 3.3. Learning Model

In this study, we use a BPNN as the learning algorithm, with a *ReLU* function on the hidden layer and sigmoid function on the output layer as activation functions. Figure 4 shows the configuration of the BPNN for smart sensing period control. The *x*_0_, *x*_1_, *x*_2_ of the *Input* layer correspond to the power consumed in a single sensing operation, the remaining power and the remaining time respectively. The output value of the y node in the *Output* layer represents a period.

In an IoT environment, as devices have a low computing power and limited resources, algorithms requiring high computational abilities cannot be implemented in the device hardware specifications. As the depth of the hidden layer in the back-propagation algorithm becomes deeper, it consumes more resources and more computing ability. Therefore, to adapt to the computing environment in IoT, it is necessary to appropriately control the depth of the hidden layer. In this paper, to decide the number of nodes in the hidden layer, repeated experiments are performed on the devices. From the experiments, we find that the appropriate number of nodes for the hidden layer is 2. When 2 nodes are applied, the learning speed is fastest. To evaluate the model, the data set created in Section 3.1 and Section 3.2 was implemented. 70% of the data set was used for the learning process and 30% of the data set was used for the testing process. After finishing the learning process, we evaluated our learning model with the test data set. As a result, our learning model was evaluated as 91% accuracy.

### 3.4. Sensing Period Control through Learning between Objects

Figure 5 shows the sequence diagram between modules participating in the smart sensing period computation. The MinT-App is an application that uses the *MinT* software. Since this App is registered as a service supported by the *MinT*, the data sensing commands from the sensor driver can be retrieved. The sensing period control is computed as follows. First, the remaining time and remaining power in the MinT-App, and the amount of consumed power in a sensing operation, are sent to the BPNN directly. Next, the elapsed time and consumed power are calculated. Second, the processes of updating the values of remaining time and power are performed. The BPNN determines a suitable sensing period using both the data from the MinT-App and the pre-designed learning model. The sensing period is sent to the Thing-Property module and the BPNN waits until the MinT-App transfers the next batch of data. The Thing-Property module is one of the modules involved in the Resource Management Layer of *MinT*. This module sets the period of sensors based on the data received from the BPNN. It also requests the sensing data from the sensors. The sensor returns the sensing data and waits until the next sensing period. After the current sensing period has elapsed, it can be updated as a new sensing period by the BBNN.

## 4. Experiment Environment

### 4.1. Experimental Devices

The experimental set-up is shown in Figure 6, it consists of a Raspberry Pi 3, a camera sensor and an ultrasonic sensor. The Raspberry Pi 3 is made by the UK-based Raspberry Foundation, and is usually chosen for IoT service program development. It operates on the *Raspbian* OS, which is based on *Debian* Linux. We build the experimental environment with the *MinT* installed on the *Raspbian*.

### 4.2. Experimental Process

To evaluate the effect of a smart sensing period, the following 3 steps are performed.

#### 4.2.1. Static Sensing Period Policy on Ultrasonic Sensor and Camera

We evaluate the effective usage of limited battery power when the static sensing period is applied. In the case of the ultrasonic sensor, the static sensing period is started at 200 ms. The sensing period increases by 200 ms in each experiment, and the maximum period value is 1000 ms; hence, we perform 5 experiments with a static sensing period. The sensing period of the camera sensor is set from 1 to 5 s, with an incremental increase of 1 s per experiment. As with the ultrasonic sensor, five experiments are performed.

#### 4.2.2. Smart Sensing Period Policy on Each Sensor

We evaluate, for the same sensors, the effective battery usage of our smart sensing period policy. The experiment is conducted on each sensor device and their lifetimes are measured.

#### 4.2.3. Smart Sensing Policy on 2 Sensors

When the two sensors mentioned above are operated simultaneously, the waste and surplus of the battery energy are measured, to verify the effects of smart sensing control. In a typical IoT working environment several sensors are attached to a sensor mode, and work together to respond to user requirements. The time limit for learning in the BPNN is 60 s. During this period, the power is reduced from 40 W to 20 W. The learning achievement according to this power change is evaluated.

## 5. Implementation and Evaluation

### 5.1. User Interface

Figure 7 shows the GUI interface for the sensing period control process, which is constructed by a Java Swing program. The user can input time and power values in the text fields marked Time and Power. The Mode button is a combo box to select a fixed or smart period policy. If the Fixed mode is chosen, the fixed value is inputted in millisecond units. Otherwise, if the Smart mode is chosen, the Period field is will be inactive and the smart sensing is started. The sensing period is controlled automatically according to the working environment. The applied sensing period values are displayed on the user screen. The sensor combo box chooses the sensor for experiment, either the ultrasonic, the camera or both. The left-hand side of Figure 7 represents the remaining battery energy and time remaining until the target time. The dialog boxes in this figure show the remaining battery, remaining time and a log record of sensing period according to the current working conditions.

### 5.2. Sensor Operation and Measurement

Figure 8 shows screenshots of the process for an ultrasonic sensor and camera sensor with fixed sensing mode. The battery and time labels in the upper window represent the current values. The print values in the bottom window represent the previous data. From Figure 8 we can confirm that the sensors are working on the sensing period established by the user.

Figure 9 shows screenshots for the ultrasonic sensor when the smart sensing mode is applied. From the log it can be seen that, after the first sensing operation is started, a constant period is maintained for a short time. However, after the initial period passes, the sensing period changes according to the remaining battery power and time. If the available battery power is much less than that required to meet the target time, the sensing period becomes longer to compensate. On the other hand, if the battery power is sufficient, the sensing period will be shortened.

Figure 10 shows screenshots of the camera sensor when the smart sensing mode is applied. Like Figure 9, the constant sensing period is for some time maintained, but then changes as time passes. As shown in this figure, the period of the camera sensor is longer than that of the ultrasonic sensor. The camera sensor requires more power than the ultrasonic sensor per sensing operation. When they are deployed in the same circumstances, the number of sensing operations of the camera sensor is limited compared to the ultrasonic one. As a result, the sensing period found from the BPNN is longer than that of the ultrasonic sensor’s.

### 5.3. Performance Evaluation

#### 5.3.1. Power Consumption in Fixed Sensing Period

Figure 11 shows the experimental results for the fixed sensing period policies. The horizontal axis represents time and the vertical axis the consumed power. In this figure, the ideal graph type is that achieved when the 20 W is completely exhausted as the 60-s time limit is reached. From Figure 11, it can be seen that the ultrasonic sensor has an ideal power consumption between 400 ms and 600 ms. When the 200 ms sensing period is applied, the energy source is exhausted in 30 s, half of the 60-s target time for this experiment. On the contrary, when a 1000 ms sensing period is applied, it uses only the half of the available power during the entire 60-s working time, and a surplus of 10 W remains.

Figure 12 shows the experimental results when a fixed sensing period is applied to the camera sensor. Since the camera sensor consumes more power per sensing operation than the ultrasonic sensor, the sensing period control is important in sustaining the longevity of the IoT network. The graph for the camera sensor takes a slanted step form. The jumps represent the increase in power consumption. The flat portions represent the waiting state until the next sensing period is reached. From Figure 12, it can be seen that, when the 1-s sensing period is applied, the total power supply is exhausted at the 25% point of the 60-s target time, the worst power consumption seen in these experiments. In addition, when the 5-s sensing period is applied, the result is only an 80% usage of total power, the largest surplus battery power seen here. In this experiment, the 4-s sensing period showed ideal power consumption results. However, this result was dependent on the specific experimental environment we used. If the conditions are varied, different results will issue.

#### 5.3.2. Power Consumption in Smart Sensing Period

Figure 13 shows the experimental results obtained when the smart sensing period is applied. The ultrasonic sensor showed the ideal power consumption graph. The camera sensor represented a case with a small amount of surplus power. The allowed error in the learning process of BPNN has an impact upon the camera sensor results. However, when compared to the ideal result of the 4-s sensing period for the fixed policy, this error is trivial.

#### 5.3.3. Smart Sensing Period on 2 Sensors

Figure 14 shows the power consumption of 2 sensors when operated simultaneously. In this figure, the camera and ultrasonic labels represent the quantity of power consumed by each sensor as the time passes. The camera + ultrasonic label represents the sum power consumption of both sensors. The figure shows that the total power (20 W) was consumed within 59 s. Even with the small power surplus remaining, the performance is improved compared to the fixed period policy results of Figure 11 and Figure 12. Several iterated experiments were necessary to find the ideal period in the case of fixed sensing period policies; however, the smart sensing period policy automatically searches out the ideal sensing period value, based on the given environmental conditions. In addition, even if both sensors are operated simultaneously, the smart sensing policy autonomously controls their power consumption. As a result, this can enable both sensors to work effectively up to the target time.

#### 5.3.4. Comparison of Efficiency

Figure 15 and Figure 16 show the power consumption rate of the ultrasonic and camera sensors. The horizontal axes represent the fixed periods and the vertical axes their power consumption rates. The power consumption rate is found from the taken for complete power consumption over total (target) time, and the remaining power over total power. It is calculated from Equation (5). In Equation (5), $T_{c}$ is the time at which all power is exhausted. *P_r_* is the power remaining from the total available power. In the case of $T_{c}$, if the total time is reached and the total power is not exhausted, it is regarded that *T_c_* is equal to total time *T*.

The power consumption rate is near to 100% when the remaining time and power are both close to 0. This result signifies a case of efficient power consumption. In these figures, when the graph for a fixed sensing period policy comes close to the graph of smart sensing one, it is regarded that the fixed sensing period policy exhausts the battery power efficiently for the given working environment. For the ultrasonic sensor, the 400 ms period is the most efficient. In the camera sensor, 4 s is the most efficient period among the fixed sensing period policies.
(5)$$\left(\frac{T_{c}}{T}-\frac{P_{r}}{P_{T}}\right)\times 100$$
$T_{c}$ = time at which all power is exhausted; $P_{r}$ = power remaining from the total available power.

Figure 17 and Figure 18 represent the error in power consumption for each sensor. The horizontal axes indicate the sensing periods and the vertical axes the power consumption error. If the available power is exactly exhausted by the target time, the power consumption error is 0. For example, the plus (+) error value indicates surplus power, and the negative (−) error value denotes an insufficient power.

From Figure 17 and Figure 18 we can analyze in detail the results of Figure 15 and Figure 16. In Figure 15, the power consumption rate for the ultrasonic sensor is highest with a 400 ms sensing period. However, we find that a negative error exists at 400 ms in Figure 17. This means that a small power wastage exists for this period. The ultrasonic sensor did not operate up to the target time at 400 ms, because the available power was completely consumed. In Figure 15, when a 1000 ms period is used, the power consumption rate is seen to be lowest. However, in Figure 17, the 1000 ms period shows a large positive error. This means that even if sensing operations continue until the target time, a considerable amount of power remains.

In Figure 15, the smart sensing period shows a power consumption rate of 100%. It indicates that a waste or surplus of power consumption did not occur in the sensing processes. This result is confirmed in Figure 17, with a 0 error value. In Figure 16, the camera sensor shows the highest power consumption rate when operating on a 4-s sensing period. As a result, an error value of 0 was recorded for the 4-s sensing period in Figure 18. Otherwise, the 1-s period in Figure 16 showed the lowest power consumption rate, and is represented by large negative error value in Figure 18. This signifies that the sensing operation was no longer supported, as the available power was completely exhausted. In this case, the extension of the current sensing period is necessary to achieve an efficient battery power usage. In Figure 16, when the camera sensor uses the smart sensing period policy, the power consumption rate was close to 100%. This was represented by a small negative error in Figure 18, so the smart sensing period policy showed a small power wastage. Despite this we can, from all experimental results, confirm that the smart sensing period showed a more efficient power consumption rate than the fixed sensing periods.

#### 5.3.5. Sensing Accuracy and Data Collection Amount

All sensors have a minimal time required for a sensing operation. If the minimal time is used as the sensing period, large quantities of sensing data are collected, and more accurate information can be obtained. The minimal times are different between sensors, but are often of the order of milliseconds or microseconds. If the minimal times of sensors are used as the sensing period, the accuracy of sensor information is increased due to the frequency of data collection. However, practically unnecessary data is stacked in the device’s internal space. In addition, the power consumption rate increases sharply. Therefore, the minimal time of sensors is not effective as the sensing period value. Despite this, even if the increase of collected data is a burden, the accurate information procured is important in IoT networks. In this paper, we use Equation (6) to evaluate the accuracy of sensing data and the battery consumption rate according to the sensing period policy. Equation (6) represents the data collection rate (*Volume Ratio*) for each sensing period policy.
(6)$$VR\left(Volume\;Ratio\right)=\frac{ST_{min}}{ST_{period}}\times 100$$
$ST_{min}$ = minimal sensing time; $ST_{period}$ = current sensing period.

In Equation (6), *ST_min_* is the minimal sensing time, and *ST_period_* the current sensing period. In the case of a fixed sensing period, the current sensing period is set as the established period value. In the case of a smart sensing period, we use the average value of applied sensing periods, as the sensing period in smart sensing policies is varied according to the working environment.

Figure 19 and Figure 20 show the data collection rates for the ultrasonic and camera sensors. The horizontal axes display the power consumption rate of the sensors, and the vertical axes indicate their data collection rates. In Figure 19, the smart sensing period is found with a 99% power consumption rate, and a 10% data collection rate. In the case of fixed sensing periods, some cases represented a higher data collection rate than the smart sensing period. However, since their power consumption rate was below 50%, the sensing operations were stopped before their target times. From these results, we can confirm that the fixed sensing periods cause a wastage or surplus in power consumption. In Figure 20, the smart sensing period recorded a power consumption rate of 95%, and a data collection rate of 21%. Otherwise, a fixed sensing period recorded data collection rates of over 80%, but the power consumption rate was below 30%. To sustain the sensing operation with a high data collection rate, the fixed sensing period requires other power supplies, instead of battery power.

## 6. Conclusions and Future Work

In this paper, a smart sensing period policy was proposed to avoid the wasted or surplus power consumption of devices in an IoT network. The proposed method uses the BPNN. It calculates the best sensing period using the learning process of the back-propagation algorithm. The learning data consisted of the remaining time, the remaining power, and the amount of power consumed in a sensing operation.

The smart sensing period was implemented with a Raspberry Pi, and we measured its performance through a three-step process. In the ultrasonic sensor experiment, the smart sensing period showed a power consumption of 99%. When compared to the fixed sensing period, its performance showed a maximum improvement of 56% for a 200 ms period, and a minimum improvement of 11% for a 400 ms period. The same experiment conducted upon the camera sensor showed a power consumption of 95%, and the performance improvement was maximized at 68% in the 1-s period. Since the smart sensing period consumed power more effectively, we confirm from these experimental results that the smart sensing period contributes to the increased longevity of devices in the IoT network.

The smart sensing period uses the limited battery power effectively until the target time is reached. However, it prolongs the sensing period according to the given working environment. This characteristic has a disadvantage in that changes occurring more frequently than the fixed sensing period cannot be measured. In IoT services, each application has its own intrinsic requirements regarding the collection of sensor data. For example, the sensing data for temperature does not change sharply within a short period, but the data required for avoiding obstacles has to satisfy real-time requirements. In the future, we will study how the smart sensing period algorithm affects the characteristics of the sensing data obtained, besides the available power and power consumption rate of each sensor.

## Figures and Tables

**Figure 1 sensors-19-04915-f001:**
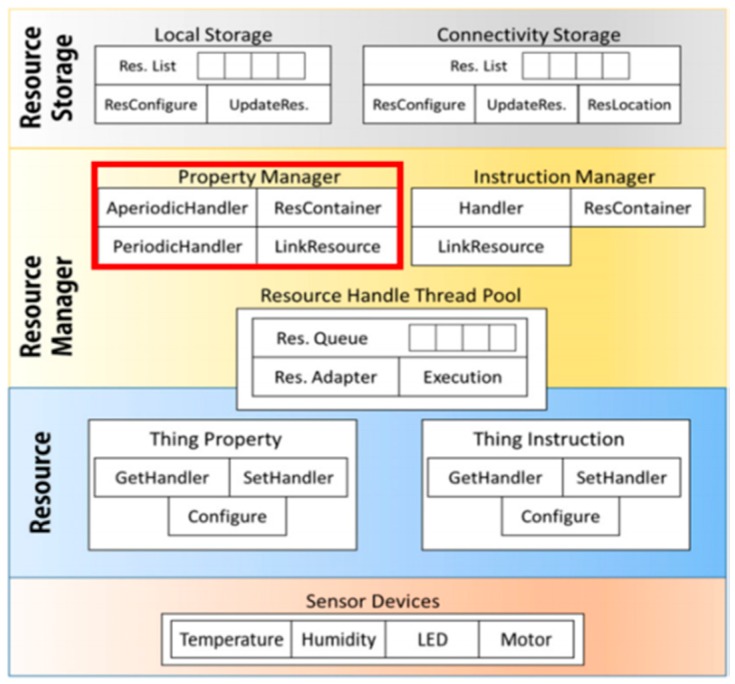
*MinT* resource management architecture. Res.: resource; LED: Light Emitting Diode.

**Figure 2 sensors-19-04915-f002:**
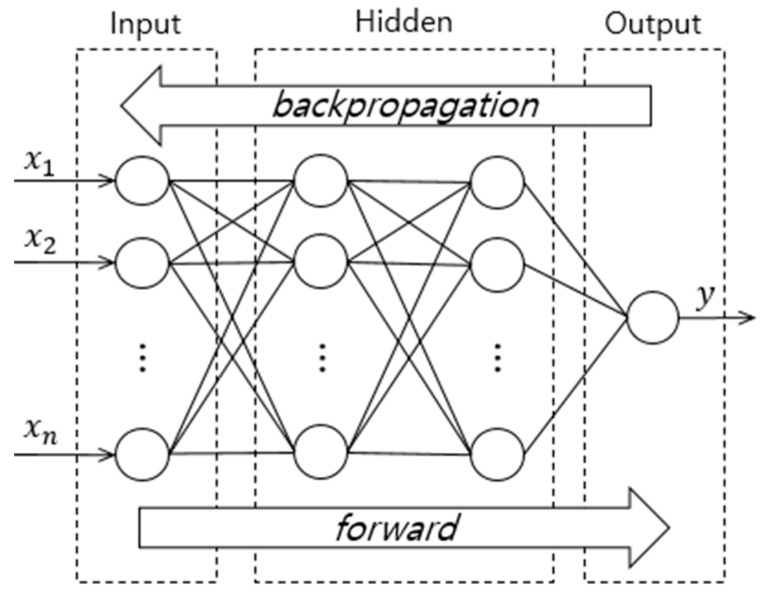
BPNN Structure.

**Figure 3 sensors-19-04915-f003:**
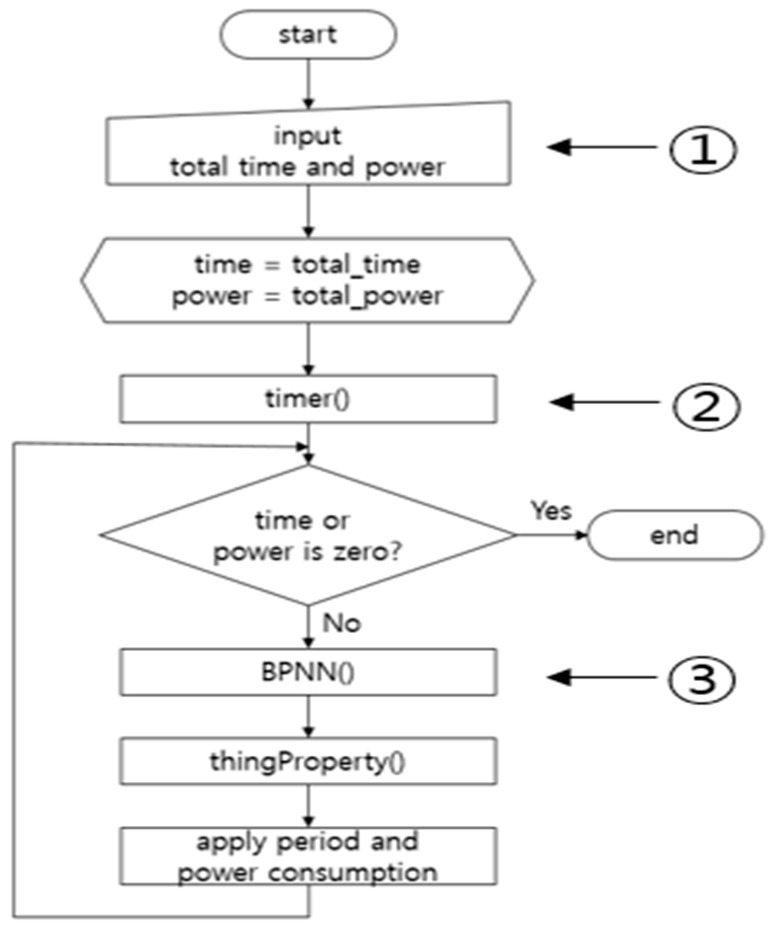
Flow chart of Smart Sensing Period.

**Figure 4 sensors-19-04915-f004:**
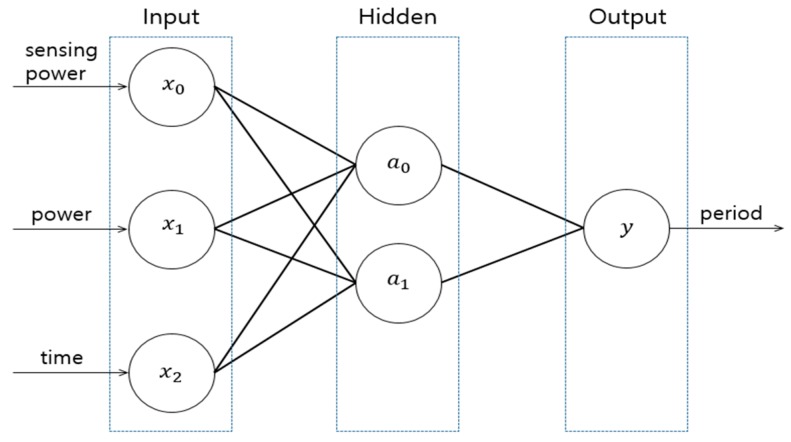
BPNN for smart sensing structures.

**Figure 5 sensors-19-04915-f005:**
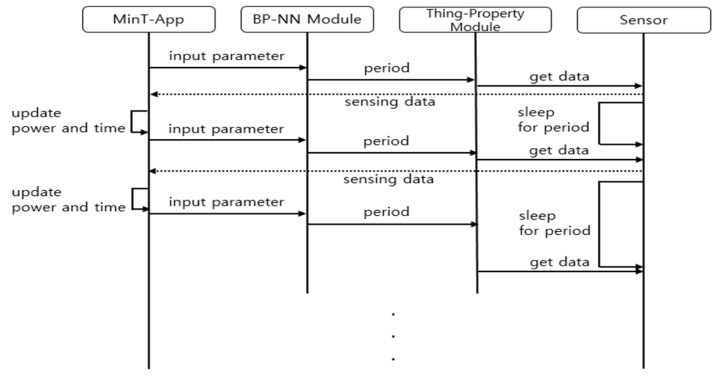
Sequence diagram for sensing period control.

**Figure 6 sensors-19-04915-f006:**
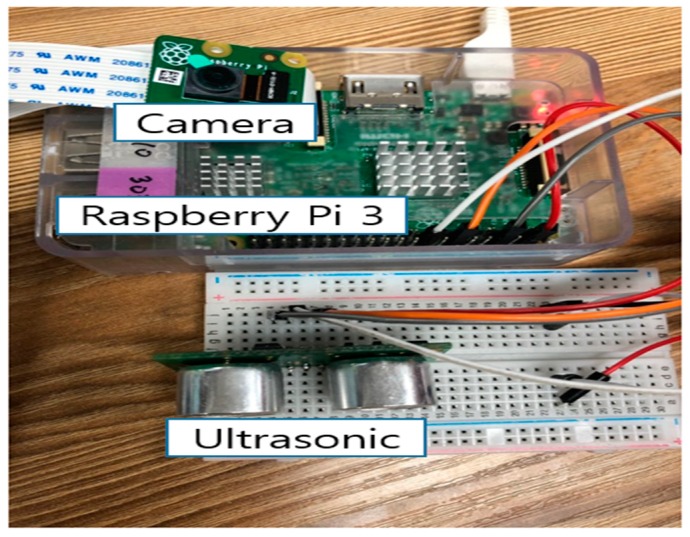
Experimental equipment (Raspberry Pi3, Ultrasonic, Camera, Raspberry Pi Foundation, Cambridge, UK).

**Figure 7 sensors-19-04915-f007:**
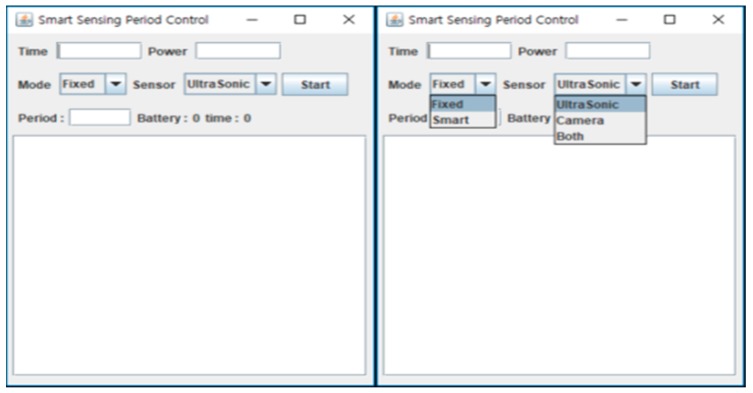
Smart sensing period control user interface.

**Figure 8 sensors-19-04915-f008:**
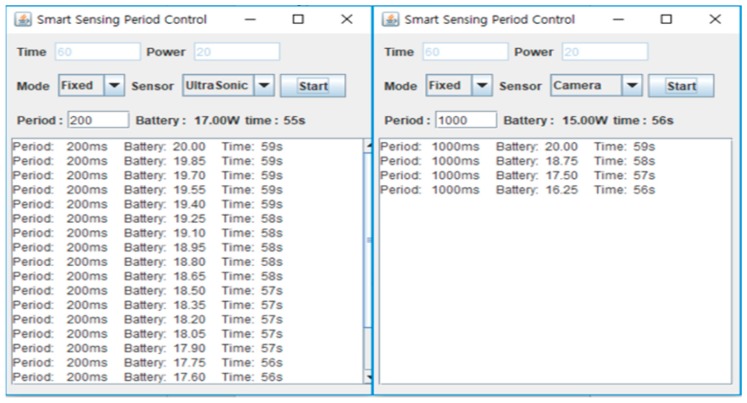
Screenshots of fixed period.

**Figure 9 sensors-19-04915-f009:**
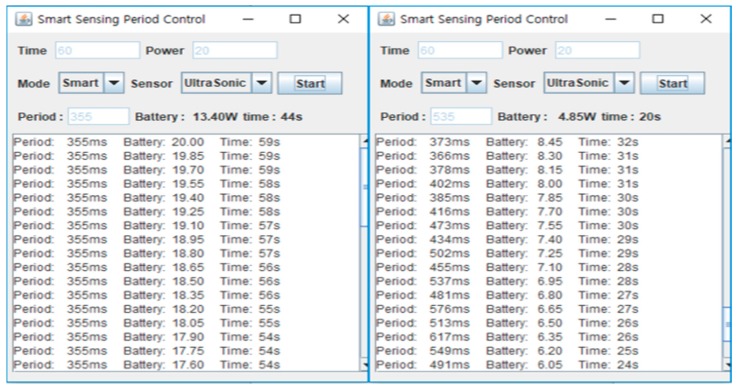
Screenshots of smart sensing in ultrasonic sensor.

**Figure 10 sensors-19-04915-f010:**
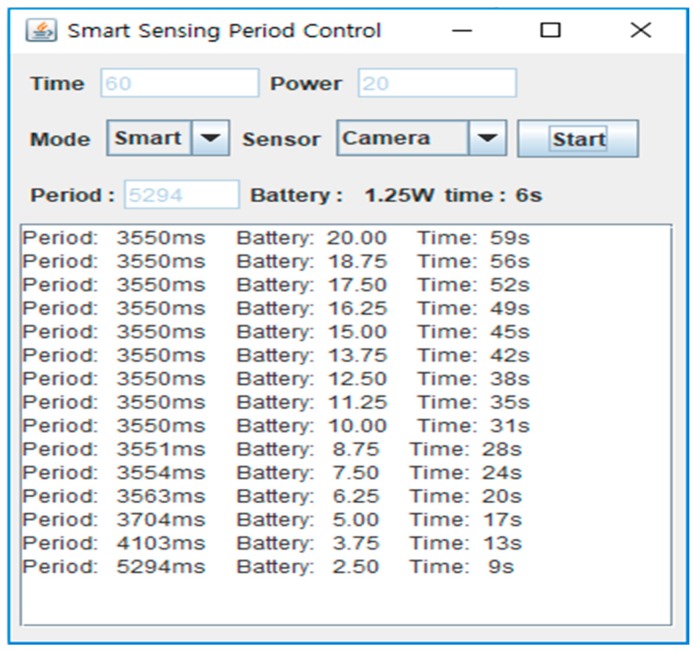
Screenshots of smart sensing in camera sensor.

**Figure 11 sensors-19-04915-f011:**
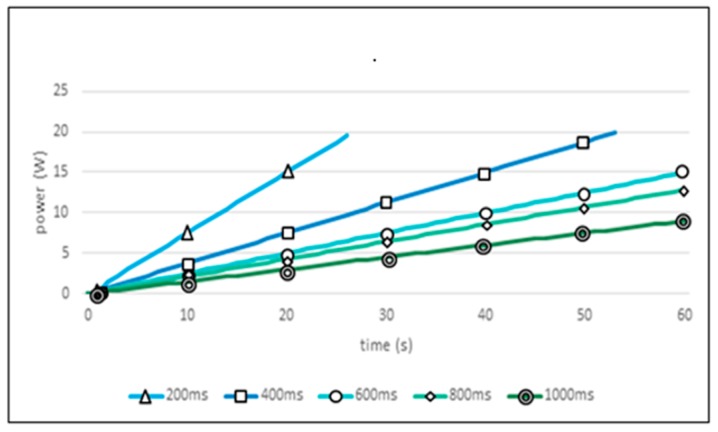
Power consumption of ultrasonic sensor in fixed sensing period.

**Figure 12 sensors-19-04915-f012:**
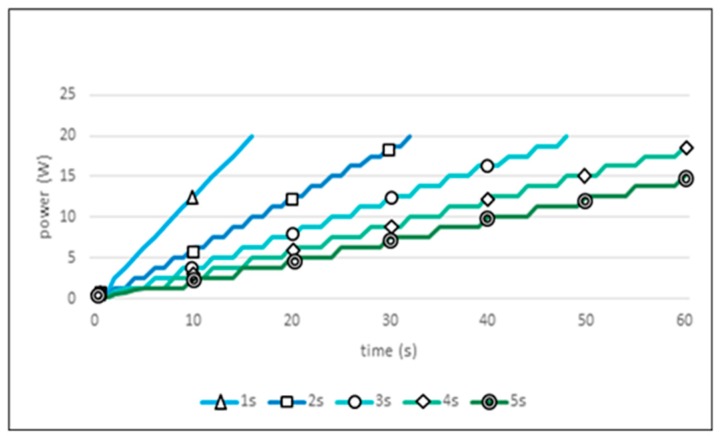
Power consumption of camera sensor in fixed sensing period.

**Figure 13 sensors-19-04915-f013:**
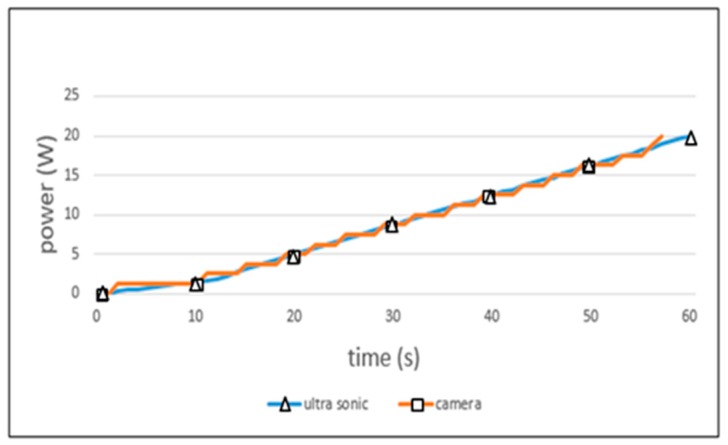
Power consumption of smart sensing period.

**Figure 14 sensors-19-04915-f014:**
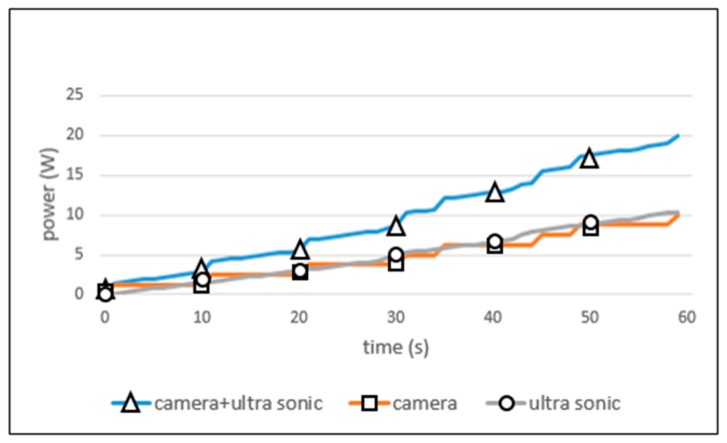
Power consumption on 2 sensors.

**Figure 15 sensors-19-04915-f015:**
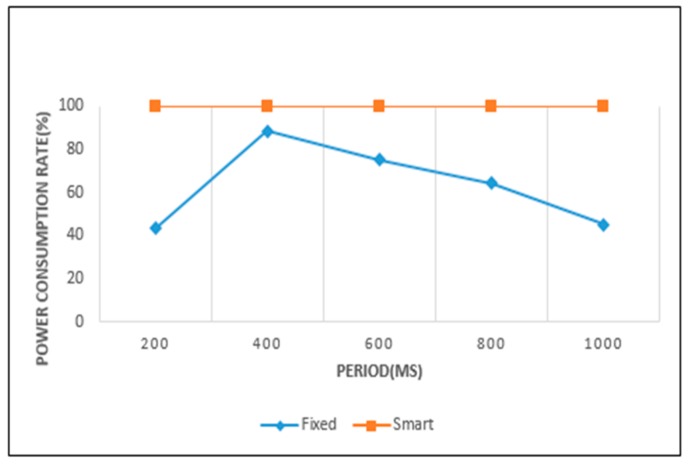
Power consumption of ultrasonic sensor.

**Figure 16 sensors-19-04915-f016:**
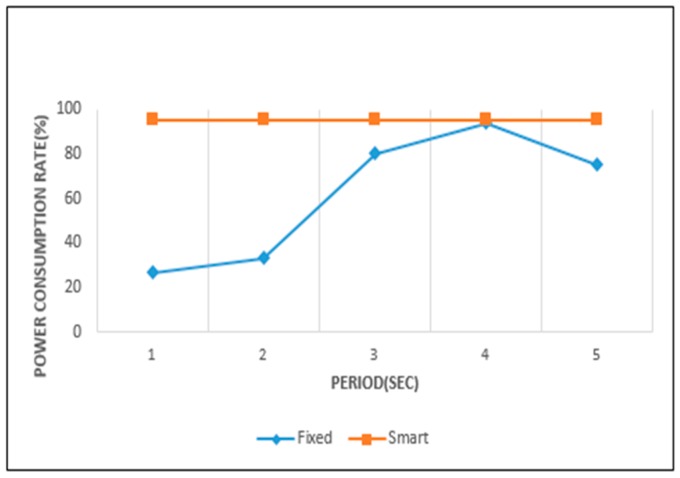
Power consumption of camera sensor.

**Figure 17 sensors-19-04915-f017:**
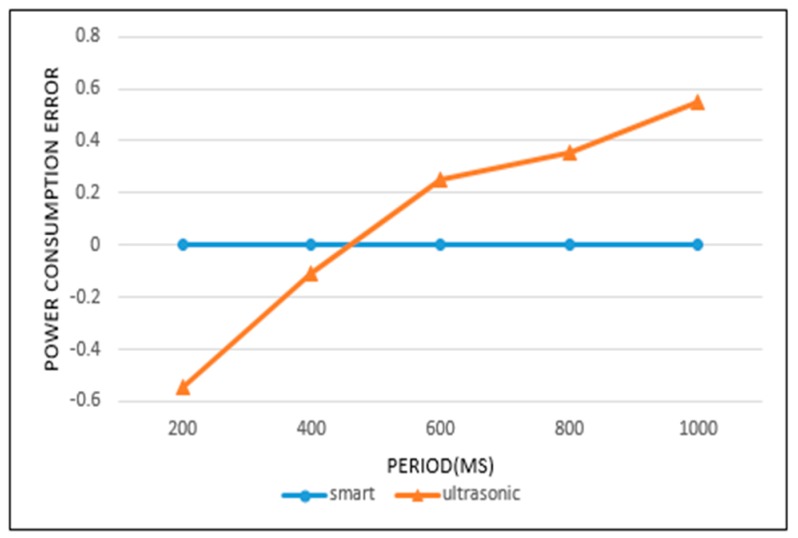
Power consumption error of ultrasonic sensor.

**Figure 18 sensors-19-04915-f018:**
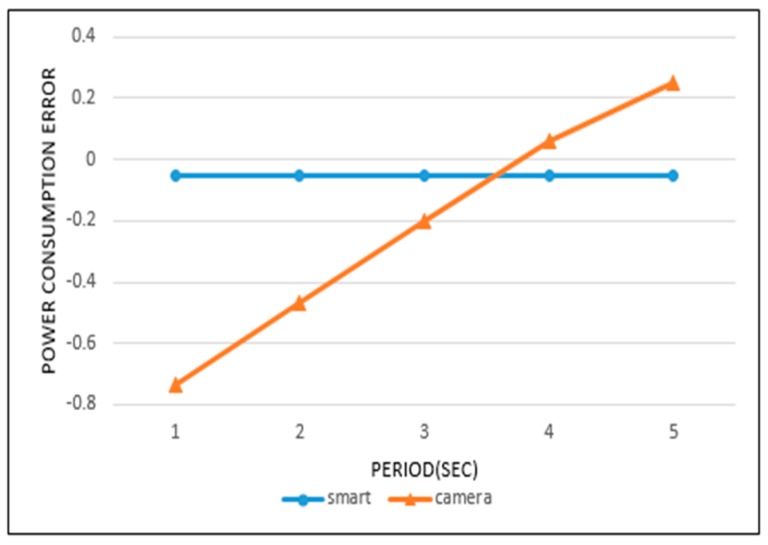
Power consumption error of camera sensor.

**Figure 19 sensors-19-04915-f019:**
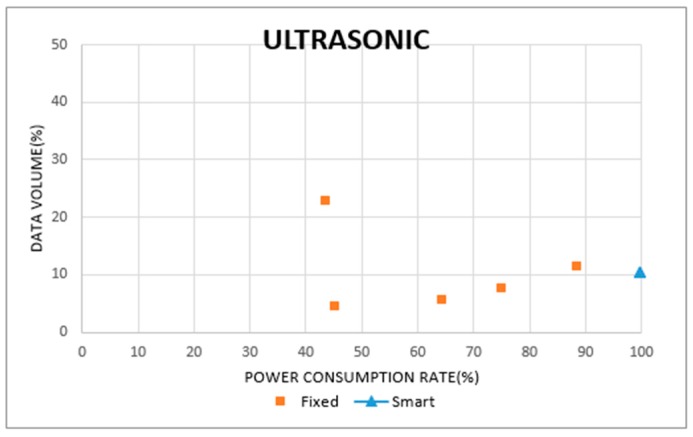
Sensing Data Volume in Ultrasonic Sensor.

**Figure 20 sensors-19-04915-f020:**
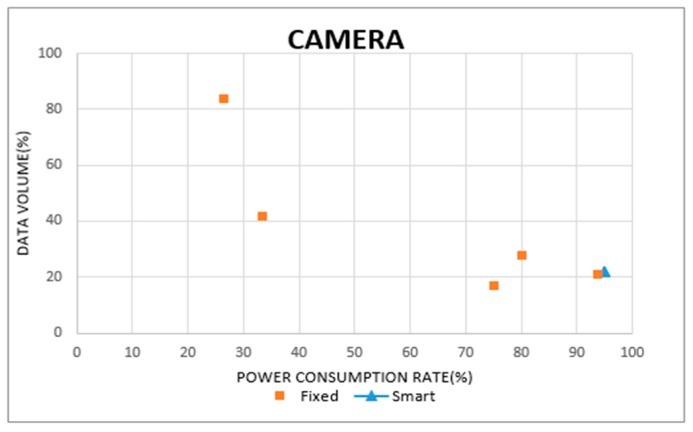
Sensing data volume in camera sensor.

**Table 1 sensors-19-04915-t001:** Part of learning data for ultrasonic sensor.

Power Consumption Per Sense Operation	Time	Power	Period (ms)
0.15	100	40.00	375.000
0.15	99	39.70	374.055
0.15	98	39.40	373.096
0.15	97	38.95	373.268
0.15	96	38.50	374.026
0.15	95	38.20	373.037
0.15	94	37.75	373.510
0.15	93	37.30	373.995

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
