# Peer review of "Smart Sensing Period for Efficient Energy Consumption in IoT Network"

_sensors, 2019, doi:10.3390/s19224915_

Round 1
Reviewer 1 Report
In this paper, a smart sensing period policy is proposed for efficient energy consumption in an IoT network. The proposed method uses a learning model based on a back-propagation neural network. Within the target time, it can efficiently use the battery energy, without any surplus or wastage in the quantity of preserved battery energy. In general, this paper is well written. The followings are some concerns
1. Fig. 1 and the figures in the simulation section is not easy to read. The fonts should be increased.
2. the equations is not clear to read, please do not use figures.
3. how about the complexity? is the neural network too heavy for the sensors?
4. The references are not enough, some references are missing and should be included
For example, as an example of Iot, "Data driven cyber-physical system for landslide detection" should be cited
Moreover, there are many other control method, such as Markov decision process and reenforcement learning, the author should cite the MDP paper (e.g. Markov-decision-process-assisted consumer scheduling in a networked smart grid ) and reenforcement learning papers (e.g. Age of Information-Aware Radio Resource Management in Vehicular Networks: A Proactive Deep Reinforcement Learning Perspective, ), and discuss the reason why using the proposed scheme in this paper. This helps the readers gain more insights.
5. It is mentioned that the comparison scheme is adopted to help evaluate the performance. Can the authors explain more about this?
Author Response
Response to Inquiries for Revision (Reviewer 1)
Title: Smart Sensing Period for Efficient Energy Consumption in IoT Network
Manuscript ID: sensors-628682
We are indebted to the reviewers for their valuable comments which were extremely helpful in revising our manuscript. We have given due considerations to each and every comment in preparing the revised manuscript.
For equation and symbol handling, we attached the word file as response.

Reviewer 2 Report
A very interesting article. Undertaking an interesting problem with interesting conclusions. However, I have a few comments. In the article, the authors presented too briefly the key issue of neural networks. Although they presented the network model, they told what algorithm they used, but there is no information about the adopted model and training data. There is also no information on how the model itself was tested as to its effectiveness. There is a lack of statistical analysis of the learning process, not even information on how convergent the learning process was.
The question arises as to how the effects obtained can be adapted to a more extensive sensor network.
The discussion of the results and presented research effects is certainly interesting. However, given that neural networks are the main research tool, more could be needed.
On pages 6 and 15, equations are illegible or difficult to read.
There is also no exact legend to the symbols used in equations.
In my opinion, corrections should also be made to English.
Author Response
Response to Inquiries for Revision (Reviewer 2)
Title: Smart Sensing Period for Efficient Energy Consumption in IoT Network
Manuscript ID: sensors-628682
We are indebted to the reviewers for their valuable comments which were extremely helpful in revising our manuscript. We have given due considerations to each and every comment in preparing the revised manuscript.
For equation and symbol handling, we attached the word file as response.

Round 2
Reviewer 1 Report
I am basically OK with this revision. One concern is that the discussion on the related references should be added to the revised paper with the suggested paper cited.
Author Response
Dear, reviewer
- We added the paper you suggested (Data driven cyber-physical system for landslide detection) in 1. Introduction and references. The content of correction explained in the appended respose file.
- We had proofreading process from professional english editing service(Editage).

Reviewer 2 Report
Ok, i agree with your answers
Author Response
Dear, reviewer
- We had proofreading process from professional english editing service(Editage).